# Controlling a One-Legged Robot to Clear Obstacles by Combining the SLIP Model with Air Trajectory Planning

**DOI:** 10.3390/biomimetics8010066

**Published:** 2023-02-05

**Authors:** Senwei Huang, Xiuli Zhang

**Affiliations:** School of Mechanical, Electronic and Control Engineering, Beijing Jiaotong University, Beijing 100044, China

**Keywords:** legged robot, SLIP, foot trajectory planning, Bézier curve, bionic hopping control

## Abstract

Legged animals can adapt to complex terrains because they can step or jump over obstacles. Their application of foot force is determined according to the estimation of the height of an obstacle; then, the trajectory of the legs is controlled to clear the obstacle. In this paper, we designed a three-DoF one-legged robot. A spring-loaded inverted pendulum model was employed to control the jumping. Herein, the jumping height was mapped to the foot force by mimicking the jumping control mechanisms of animals. The foot trajectory in the air was planned using the Bézier curve. Finally, the experiments of the one-legged robot jumping over multiple obstacles of different heights were implemented in the PyBullet simulation environment. The simulation results demonstrate the effectiveness of the method proposed in this paper.

## 1. Introduction

Legged robots are a research hotspot in the field of robotics and have achieved excellent performance in structural design and motion control [1,2,3,4,5,6,7,8,9,10]. The one-legged robot is a special kind of legged robot because it is the basis of legged robot research. Most technologies of multi-legged robots are extended from one-legged robots. Therefore, the breakthrough of one-legged-robot control technology is the key to promoting the development of legged robots. Research on the hopping control of single-legged robots has been conducted for more than 40 years and has also yielded excellent research results. Raibert decomposed the hopping process into three parts and realized the continuous hopping of a physical one-legged robot [11]. Inspired by the structure of human joints, a one-legged robot with a controllable reduction ratio was created, achieving a 42 cm vertical jump [12]. Salto-1P, an untethered monopedal robot, can hop on discontinuous surfaces such as office furniture and track a moving platform. Salto-1P’s deadbeat foot placement jumping controller allows it to precisely position the foot even on trajectories with abrupt changes in height, speed, and direction [13]. A parallel one-legged robot with a brushless direct current motor and a harmonic driver uses a virtual model control (VMC) approach to jump and land with high dynamic motion [14]. Moreover, the efficiency of virtual model control has been validated for one-legged robots [15,16]. There are other ways to achieve legged robot hopping, such as ZMP [17,18], MPC [19,20], WBC [21,22,23], and trajectory optimization-based methods [24,25,26]. Moreover, artificial intelligence algorithms, such as reinforcement learning and deep reinforcement learning, are also applied to jumping control in legged robots [27,28]. In these works, the researchers focused on the structural design, jumping height, running speed, and stability control of the one-legged robot. The performance of various control methods is also closely related to the robot structure and control objectives. In addition, some approaches need to be extended to the full locomotion task. The advantage of a one-legged robot is its exceptional ability to jump and clear obstacles. However, few studies have covered the obstacle clearing of the one-legged robot. On the other hand, trajectory planning is also extremely important for one-legged robots and has been ignored by researchers.

Jumping is an important locomotion strategy employed by animals such as frogs, kangaroos, and galagos. Compared with walking and running, jumping can better adapt animals to unstructured environments and even allow them to overcome obstacles several times larger than their body size [29,30]. By using mathematical models to study the muscle properties, leg conformation, and jumping technique of humans, frogs, locusts, and other animals during jumping, it was found that jump height increases with the increase in isometric force exerted by the leg muscles, maximum shortening velocity, and a series of consistencies [31]. Animals can predict their jumping range according to their weight, leg length, and the amount of energy they can generate, so as to assess whether they can cross obstacles in the environment [32]. Legged animals such as dogs, kangaroos, and locusts plan their foot trajectory and jump height according to the shape of the obstacle [33,34]. The body attitude of the froghopper in jumping is set by the front and middle legs, and the attitude is adjusted by the extension of the hind legs to control the direction of movement after take-off [35]. Humans can cross obstacles by controlling the trajectory of their bodies as well as the amount of foot force. Furthermore, animals can perceive ground information through the visual and tactile systems and then plan movement trajectories according to the size and location of obstacles [36,37]. The animal control of body trajectory and foot force is what the robot should learn in order to improve motion performance. Engineering practice has also found that the robot finds it difficult to walk over obstacles with compliant behaviors, because certain motions such as the lifting of feet are needed. However, environment perception and trajectory planning are important problems that jumping robots still have to solve before they can leave the laboratory and be applied in real unstructured environments. Therefore, one research focus in bionic legged robots is to explore the jumping mechanisms of animals and imitate their behavior. Biologists have summarized the dynamic motion of animals using the spring-loaded inverted pendulum (SLIP) model, which is commonly used in hopping control in legged robots [38]. In the SLIP model, the mass of the robot is centralized in the center of the body, and the legs act as a massless spring. The jumping mechanism of locusts well reflects the SLIP model, where the movement of the locust legs during jumping acts like a spring to store and release energy [39]. Moreover, biological research has revealed that the stiffness of animal legs changes dynamically [40,41]. This also explains why it is difficult to obtain good performance when using fixed SLIP model spring stiffness to describe the dynamic characteristics of robot legs. To address this issue, methods for qualitatively analyzing the change in spring stiffness with the increase in running speed have been proposed [42,43].

In this paper, we present an obstacle-clearing method for a bio-inspired one-legged hopping robot that combines the SLIP model with air trajectory planning. The inspiration comes from planning the body trajectory and the magnitude of the foot force according to the height and shape of the obstacles when a human or an animal performs a high jump. To verify the effectiveness of the proposed control method and estimate the jumping performance of the one-legged robot, a simulation experiment was designed to control a one-legged robot continuously jumping over several obstacles of different heights. The experimental results show that the controller designed in this paper can realize one-legged-robot jumping and obstacle-clearing control, which verifies that the control strategy proposed in this paper is effective. The novelty and main contribution of this paper are to propose a method for incorporating Bézier curve-based foot trajectory planning into SLIP model control to imitate the jumping and obstacle-clearing mechanism of animals. It also proposes a method to supplement the energy loss of the robot according to the relationship between the foot force and the speed of the animal when running.

The remainder of the paper is organized as follows: In Section 2, the one-legged robot structure and kinematics are presented. Section 3 introduces foot trajectory planning based on the Bézier curve. Section 4 introduces the control method and reports the building procedure of the controller of the one-legged robot. Section 5 presents the obstacle-clearing experiment and analyzes the results. Finally, Section 6 concludes the paper and discusses future work.

## 2. One-Legged Robot

### 2.1. Structural Design

By mimicking the legs of quadrupeds, we built a one-legged robot using PyBullet simulation software. The robot was composed of a body, a femur, a tibia, and a foot, as shown in Figure 1a. The robot had three degrees of freedom: hip roll, hip pitch, and knee pitch. The foot was fixed at the lower end of the tibia. The model parameters are listed in Table 1. The robot’s dimensions were 0.3 m × 0.3 m × 0.8 m, and it weighed 130 kg.

### 2.2. Kinematic Model

The coordinate frames of the one-legged robot are shown in Figure 1b. {W} is the world coordinate frame with the origin fixed at the center of the ground. {B} is the body coordinate frame with the origin fixed at the geometric center of the body. {H} is the reference coordinate frame for foot trajectory planning, with the same origin as {B} and the axes always oriented in the same direction as {W}. {hr} and {hp} are the hip joint roll and pitch coordinate frame, respectively, with the origin fixed at the geometric center of the hip joint. {kp} is the knee joint pitch coordinate frame with the origin fixed at the geometric center of the knee joint. {f} is the foot coordinate frame with the origin fixed at the geometric center of the foot. The x-axis of all coordinate frames points in the direction of robot movement; the z-axis of all coordinate frames is vertically upward-oriented; and the y-axis of all coordinate frames is determined according to the right-hand rule. The kinematics model of the one-legged robot was established based on the D-H method. According to the forward kinematics, the position of the foot in the {B} coordinate frame can be represented as
(1)xfB=l1c2+l2c23yfB=l1s2+l2s23s1zfB=−l1s2+l2s23c1
where $s_{i}=\sin\left(\theta_{i}\right)$, $c_{i}=\cos\left(\theta_{i}\right)$, $s_{ij}=\sin\left(\theta_{i}+\theta_{j}\right)$, $c_{ij}=\cos\left(\theta_{i}+\theta_{j}\right)$, and i,j=1,2,3 i≠j.

The inverse kinematics are
(2)θ1=−arctanyfBzfBθ2=arctanl1+c3l2s3l2−arctanxfB±lvir2−xf2Bθ3=arccoslvir2−l12−l222l2l3
where lvir=xf2B+yf2B+zf2B is the virtual leg length.

The partial differential of Equation (1) is calculated to obtain the foot Jacobian matrix from the foot to the body coordinate frame as follows:(3)$$J=\left[\begin{array}{ccc} 0 & -l_{1}s_{2}-l_{2}s_{23} & -l_{2}s_{23}\\ c_{1}\left(l_{1}s_{2}+l_{2}s_{23}\right) & s_{1}\left(l_{1}c_{2}+l_{2}c_{23}\right) & l_{2}s_{1}c_{23}\\ s_{1}\left(l_{1}s_{2}+l_{2}s_{23}\right) & -c_{1}\left(l_{1}c_{2}+l_{2}c_{23}\right) & -l_{2}c_{1}c_{23} \end{array}\right]$$

## 3. Foot Trajectory Planning Based on the Bézier Curve

There are numerous methods for planning foot trajectory, such as the rectangular planning method [44], the elliptical planning method [45], the composite cycloid-based approach [46], the composite trajectory planning method [47], the polynomial-based method [48], and the Bézier curve-based approach [49]. Compared with other foot trajectory planning methods, the Bézier curve-based method has the following advantages: (1) the Bézier curve changes smoothly from the starting point to the end point due to its continuous higher-order derivatives, which can ensure continuous and smooth motion; (2) the derivative of the n-order Bézier curve is the n-1-order Bézier curve, so the velocity and acceleration of the foot can be planned; (3) the shape of the curve is determined by a relatively small number of control points; (4) because the first and last control points of the curve are on the curve, it is possible to splice multiple Bézier curves to achieve a bionic foot trajectory with more diverse shapes. Therefore, this paper uses the Bézier curve to generate and parameterize the foot trajectory of the swing phase. A Bézier curve of degree *n* can be represented as
(4)$$C(u)=\sum_{i=0}^{n}B_{n,i}(u)P_{i}$$
where $B_{n,i}(u)$ is a Bernstein polynomial as defined by $B_{n,i}(u)=\frac{n!}{i!(n-i)!}u^{i}{\left(1-u\right)}^{n-i}$, u∈[0,1] is the Bézier curve parameter, and $P_{i}$ are the control points.

The time parameter is introduced to organize the time of the foot trajectory.
(5)$$u=at,\;t\in [0,t_{\max}]$$
where $t_{\max}$ is the time it takes the foot to move from the start to the end of the trajectory and a is the time coefficient given by $a=1/t_{\max}$.

The speed of foot movement can be obtained by calculating the first derivative of the foot trajectory.
(6)$$\dot{C}(at)=\sum_{i=0}^{n-1}B_{n-1,i}(at)\left[na(P_{i+1}-P_{i})\right]$$

The acceleration of foot movement can be obtained by calculating the second derivative of the foot trajectory.
(7)$$\ddot{C}(at)=\sum_{i=0}^{n-2}B_{n-2,i}(at)\left[n\left(n-1\right)a^{2}(P_{i+2}-2P_{i+1}+P_{i})\right]$$

According to Equations (4), (6), and (7), it is simple to acquire
(8)$$\left\{\begin{array}{l} C(0)=P_{0}\\ \dot{C}\left(0\right)=na\left(P_{1}-P_{0}\right)\\ \ddot{C}\left(0\right)=na\left(P_{2}-2P_{1}+P_{0}\right) \end{array}\right.,\;t=0$$
(9)$$\left\{\begin{array}{l} C(1)=P_{n}\\ \dot{C}(1)=na(P_{n}-P_{n-1})\\ \ddot{C}(1)=n\left(n-1\right)a^{2}(P_{n}-2P_{n-1}+P_{n-2}) \end{array}\right.,\;t=t_{\max}$$

The trajectory beginning and ending points correspond to the first and last control points, respectively. The positions of the first two control points and the last two control points, and the time coefficient can be used to control the speed of the trajectory start and end points, respectively. The positions of the first three control points and the last three control points, and the time coefficient can be used to control the acceleration of the trajectory start and end points, respectively.

To make the foot trajectory smoother and avoid having too many parameters, the foot trajectory of the swing phase is determined by a fifth-order Bézier curve. The foot trajectory is planned in the {H} frame. The coordinates of the foot trajectory control points are given by
(10)PfH=xtoxtoxtoxtdxtdxtdytoytoytoytdytdytdztoztozto+hfootztd+hfootztdztd
where $\left(x_{\mathrm{to}},y_{\mathrm{to}},z_{\mathrm{to}}\right)$ is the foot position at take-off, $\left(x_{\mathrm{td}},y_{\mathrm{td}},z_{\mathrm{td}}\right)$ is the desired foot position at touch-down, and $h_{\mathrm{foot}}$ is used to control the maximum ground clearance, which is determined by the obstacle height.

## 4. Jumping Control Based on SLIP Model

The robot is equivalent to a SLIP model. The virtual spring connects the hip joint to the foot, which is called the virtual leg. The virtual leg can mimic the spring motion by controlling the pitch angle of the knee joint. The jumping process is decoupled into jumping height control, horizontal speed control, and body attitude control using the method proposed by Raibert [11]. In the stance phase, the jumping height and body attitude are controlled by adjusting the ground reaction force. In the flight phase, the horizontal speed of the next jump is changed by controlling the position of the foot.

### 4.1. Jumping Height Control

In the stance phase, the virtual spring force is adjusted to change the ground reaction force on the foot, in order to regulate the jumping height. The virtual spring force can be expressed as
(11)$$F_{\mathrm{vir}}=k_{\mathrm{ss}}\left(l_{\mathrm{vir}}^{d}-l_{\mathrm{vir}}\right)+k_{\mathrm{ds}}{\dot{l}}_{\mathrm{vir}}$$
where $k_{\mathrm{ss}}$ is the virtual spring stiffness, $k_{\mathrm{ds}}$ is the virtual spring damping, $l_{\mathrm{vir}}^{d}$ is the original length of the virtual spring, and $l_{\mathrm{vir}}$ is the virtual spring length.

To account for the system energy loss caused by friction and damping, extra force is added to the spring during the thrust phase of the stance phase. These energy losses vary with speed, and it is difficult to replenish energy manually in real time to maintain the stable jumping of the robot. Cheetahs and greyhounds increase their foot force with the increase in speed when running and adjust their jumping height by changing the vertical impulse of the stance phase [50]. The logarithmic curve is used to fit the correlation between the peak foot force and the forward speed as the basis for energy supplements. Therefore, the added force is defined as
(12)$$F_{\mathrm{add}}=\alpha \log_{a}\left(\dot{x}+1\right)$$
where $\dot{x}$ is the forward speed; a=2.779 is the logarithmic base determined by fitting; and α is the amplitude of added force, which is determined by the characteristics of the system, including mass, damping, and friction.

Then, the virtual spring force can be rewritten as
(13)$$F_{\mathrm{vir}}=\left\{\begin{array}{l} k_{\mathrm{ss}}\left(l_{\mathrm{vir}}^{d}-l_{\mathrm{vir}}\right)+k_{\mathrm{ds}}{\dot{l}}_{\mathrm{vir}},\;\mathrm{at}\;\mathrm{compression}\;\mathrm{phase}\\ k_{\mathrm{ss}}\left(l_{\mathrm{vir}}^{d}-l_{\mathrm{vir}}\right)+k_{\mathrm{ds}}{\dot{l}}_{\mathrm{vir}}+F_{\mathrm{add}},\;\mathrm{at}\;\mathrm{thrust}\;\mathrm{phase} \end{array}\right.$$

The virtual spring force calculated with Equation (13) is regarded as the force on the foot. Therefore, according to the geometric relationship, the foot output force can be expressed as
(14)$$F_{\mathrm{foot}}=\left[\begin{array}{c} \cos\theta_{\mathrm{vir}}\\ -\sin\theta_{\mathrm{vir}}\sin\theta_{1}\\ -\sin\theta_{\mathrm{vir}}\cos\theta_{1} \end{array}\right]F_{\mathrm{vir}}$$
where $\theta_{\mathrm{vir}}=\theta_{2}+\arcsin\left(l_{2}\sin\theta_{3}/l_{\mathrm{vir}}\right)$ is the virtual leg pitch angle.

The foot Jacobian matrix is used to convert the foot output force to the needed equivalent joint torque.
(15)$$\tau =J^{T}F_{\mathrm{foot}}$$
where $\tau =\left[\begin{array}{ccc} \tau_{\mathrm{hr}} & \tau_{\mathrm{hp}} & \tau_{\mathrm{kp}} \end{array}\right]$ and each component represents the torque of hip roll, hip pitch, and knee pitch, respectively.

### 4.2. Horizontal Speed Control

The horizontal speed can be controlled by adjusting the position of the foot relative to the neutral point, as shown in Figure 2. If the foot is placed on the right side of the neutral point, that is, $x_{f_{0}}>\dot{x}T_{s}/2$, part of the kinetic energy is transformed into gravitational potential energy in the stance phase, so the robot’s speed decreases, that is, ${\dot{x}}_{i}>{\dot{x}}_{i+1}$. If the foot is placed at the neutral point, that is, $x_{f_{0}}=\dot{x}T_{s}/2$, the potential energy at touch-down and take-off is the same, so the robot’s speed remains unchanged, that is, ${\dot{x}}_{i}={\dot{x}}_{i+1}$. If the foot is placed on the left side of the neutral point, that is, $x_{f_{0}}<\dot{x}T_{s}/2$, part of the gravitational potential energy is transformed into kinetic energy in the stance phase, so the robot’s speed increases, that is, ${\dot{x}}_{i}<{\dot{x}}_{i+1}$.

The acceleration and deceleration control of the robot can be realized by adding an offset $x_{f\Delta }$ on the basis that the foot is placed at the neutral point. The linear estimation of horizontal velocity is used to determine the offset.
(16)$$x_{f\Delta }=k_{\dot{x}}(\dot{x}-{\dot{x}}_{d})$$
where ${\dot{x}}_{d}$ is the expected horizontal velocity and $k_{\dot{x}}$ is the speed increment coefficient, which determines how fast the actual speed can reach the desired speed.

To sum up, the expected foot placement at an expected horizontal speed is defined as
(17)$$x_{f}=\frac{\dot{x}T_{s}}{2}+k_{\dot{x}}(\dot{x}-{\dot{x}}_{d})$$

According to the inverse kinematics Equation (2), the desired hip joint angle, $\theta^{d}$, can be obtained. Proportional differential (PD) control is adopted to obtain the hip joint torque to place the foot in the desired position:(18)$$\tau_{i}=k_{\mathrm{pf}}\left(\theta_{i}-\theta_{i}^{d}\right)-k_{\mathrm{df}}{\dot{\theta }}_{i},\;i=\mathrm{hr},\mathrm{hp}$$
where $k_{\mathrm{pf}}$ and $k_{\mathrm{df}}$ are the position and velocity gain of foot placement, respectively.

### 4.3. Body Attitude Control

In the stance phase, the body attitude is adjusted by controlling the hip joint torque. The expected body attitude angle is 0. The body attitude adjustment torque is calculated using PD control.
(19)$$\left\{\begin{array}{l} \tau_{\mathrm{hr}}=-k_{\mathrm{pb}}\theta_{\mathrm{br}}-k_{\mathrm{db}}{\dot{\theta }}_{\mathrm{br}}\\ \tau_{\mathrm{hp}}=-k_{\mathrm{pb}}\theta_{\mathrm{bp}}-k_{\mathrm{db}}{\dot{\theta }}_{\mathrm{bp}} \end{array}\right.$$
where $\theta_{\mathrm{br}}$ and $\theta_{\mathrm{bp}}$ are the body roll and pitch angles, and $k_{\mathrm{pb}}$ and $k_{\mathrm{db}}$ are the proportional and derivative gains of body attitude, respectively.

### 4.4. Gravity Compensation

The SLIP model assumes that the mass of the robot is concentrated at the center of mass of the body and ignores the mass of the leg, so that the leg motion does not influence the overall motion of the robot. However, for the leg mechanism in this paper, the gravity of the leg link always has a momentary effect on each joint. Neglecting the mass of the leg would affect the stability of the robot, so gravity compensation is required. The dynamic equation of the robot established using the Lagrange method shows that the moment generated by gravity on each joint is
(20)$$G=\left[\begin{array}{c} \frac{1}{2}s_{1}s_{2}l_{1}m_{1}g+\left(l_{1}s_{1}s_{2}+\frac{1}{2}l_{2}s_{1}s_{23}\right)m_{2}g\\ -\frac{1}{2}c_{1}c_{2}l_{1}m_{1}g-\left(l_{1}c_{1}c_{2}+\frac{1}{2}l_{2}c_{1}c_{23}\right)m_{2}g\\ -\frac{1}{2}c_{1}c_{23}l_{2}m_{2}g \end{array}\right]$$

The torque to be compensated for each joint is
(21)τ=−G

### 4.5. Finite-State Machine

To impose the jumping motion engendered from the plan, a finite-state machine (FSM) is established, as shown in Table 2. The robot goes through one stance phase, in which the foot touches the ground, and one flight phase, in which the foot does not touch the ground. The stance phase is split into two sub-phases: compression and thrust. The flight phase is split into two sub-phases: swing and landing. Switching from or to the stance phase occurs when the leg leaves or hits the ground. The transition from or to the stance phase arises when the leg leaves or touches the ground. The FSM executes the specified action in the specified state.

### 4.6. Controller of One-Legged Hopping Robot

By combining the algorithms described in this section, a stable one-legged-robot jumping motion controller is provided, as shown in Figure 3. The FSM module is responsible for coordinating the execution of actions in different states. The states of the robot are introduced into each module through feedback to form a closed control loop.

## 5. Jump Simulation Experiment

To examine the performance of the one-legged robot and designed controller, a continuous hopping experiment was carried out. The one-legged-robot model and experimental environment were built in the PyBullet simulation environment. The goal of the robot was to continuously clear obstacles of different heights in the simulation environment. Five obstacles were placed at an interval of 4 m, and the height increased from 0.1 m to 0.5 m. The control parameters set in the simulation environment using parameter tuning are listed in Table 3. Since there was no control method applied to the yaw direction movement, the robot was constrained to moving in a straight line.

In the experiment, the robot jumped forward steadily for a distance of 25 m within 25 s and continuously cleared five walls varying in height from 0.1 to 0.5 m, as shown in Figure 4. The distance between the walls was 4 m, leaving enough distance for the robot to adjust the jumping speed and height to clear the next obstacle. Figure 5 shows the change in body position. The maximum displacement of the body in the vertical direction was 1.4 m, which means that the maximum jump height of the robot reached 0.6 m. In addition, the jumping height could also be enhanced by improving the added foot force.

The velocity of the robot is shown in Figure 6. The robot achieved a forward speed of about 1 m/s at the expected speed of 1.5 m/s. The failure to achieve the desired velocity could be due to the robot tipping over if the foot placement deviated too much from the center of mass. In the meantime, the forward speed fluctuated due to the inaccuracy of foot placement estimation, so the robot constantly adjusted its forward speed during jumping. After entering the stable jump, the vertical speed continued to change steadily as the robot fell and rose.

The desired and actual foot trajectories in the swing phase were nearly consistent, as shown in Figure 7. This indicates that, after entering the swing phase, the controller could quickly plan and follow the foot trajectory according to the position of the foot placement and the ground clearance. Before crossing the obstacle, the robot first lifted its leg, and then set it down after crossing the obstacle, which is similar to humans stepping over steps. In the stance phase, the foot is expected to remain in contact with the ground. There was a period in each jump cycle with the actual foot position remaining 0 in the z-direction and constant in the x-direction, indicating that the foot in the stance phase neither rebounded nor slipped.

Figure 8 shows the three joint torques within 5 s. Jumping is a type of movement in which take-off and touch-down occur alternately. The interaction force between the robot and the external environment changes dramatically as a result of the impact generation and conversion of kinetic and potential energy during the process of touch-down and take-off. As a result, the torque exerted on joints changed periodically throughout the moving process and fluctuated greatly in the stance phase. The torque of the hip pitch was the largest, probably due to the greater pitching motion of the forward jump. As the motion was constrained in the sagittal plane, the torque of the roll hip joint was kept at 0.

## 6. Discussion and Conclusions

In this study, we demonstrate a novel idea in the research area of one-legged-robot control, which applies the Bézier curve to planning foot trajectory and combines the SLIP model to clear obstacles. The foot trajectory of the swing phase is parameterized and generated using a fifth-order Bézier curve. Moreover, the foot force is associated with speed by imitating the running mechanism of the cheetah. Finally, the one-legged robot achieved stable jumping and continuously cleared obstacles of different heights in the simulated experiment, indicating that the control method is effective. In future work, we plan to enlarge the control scheme to achieve omnidirectional motion control. In addition, we plan to transfer the trajectory planning into a nonlinear optimization by applying GA, PSO, and other intelligent optimization methods. In follow-up works, we also plan to train the parameters of the controller based on a reinforcement learning algorithm to achieve a more stable jump.

## Figures and Tables

**Figure 1 biomimetics-08-00066-f001:**
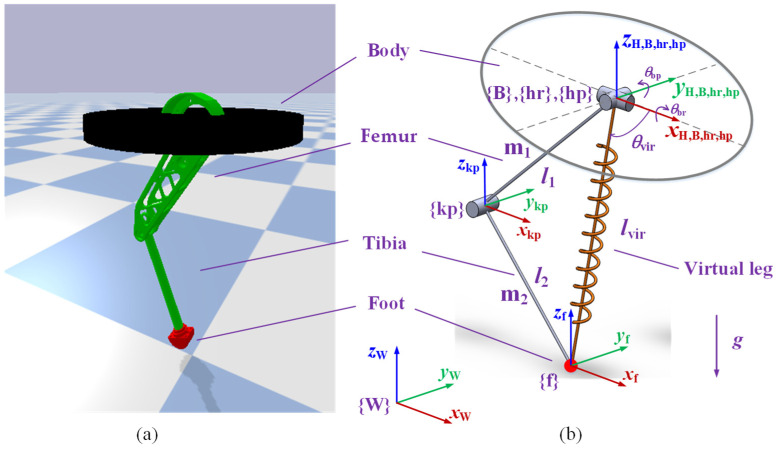
Model of the one-legged robot. (**a**) Three-dimensional model. (**b**) Kinematic model.

**Figure 2 biomimetics-08-00066-f002:**
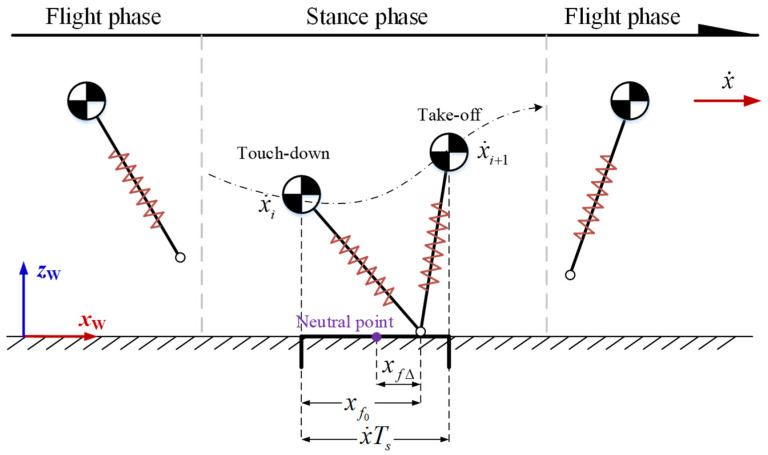
Foot placement and horizontal speed control. $x_{f_{0}}$ is the forward displacement of the foot relative to the center of mass; $\dot{x}$ is the forward horizontal velocity of the center of mass; and $T_{s}$ is the duration of the stance phase.

**Figure 3 biomimetics-08-00066-f003:**
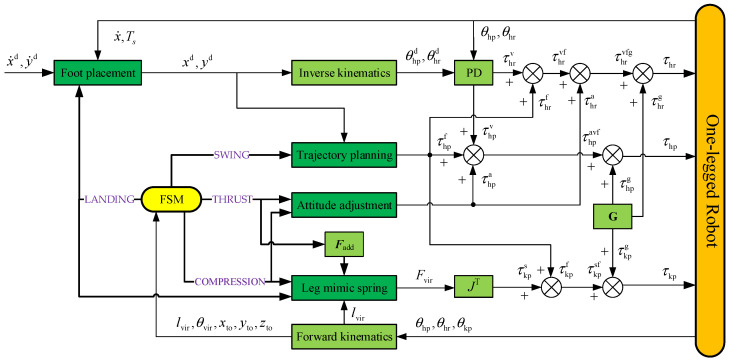
One-legged-robot controller.

**Figure 4 biomimetics-08-00066-f004:**
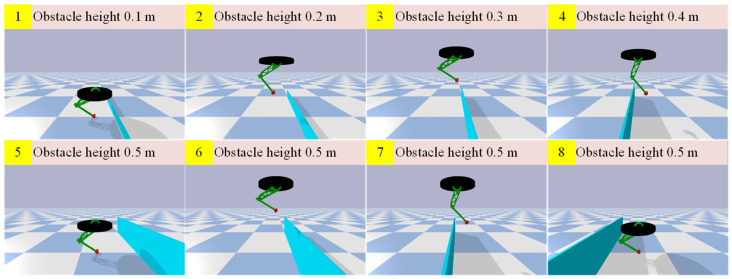
The one-legged robot continuously clears obstacles of different heights.

**Figure 5 biomimetics-08-00066-f005:**
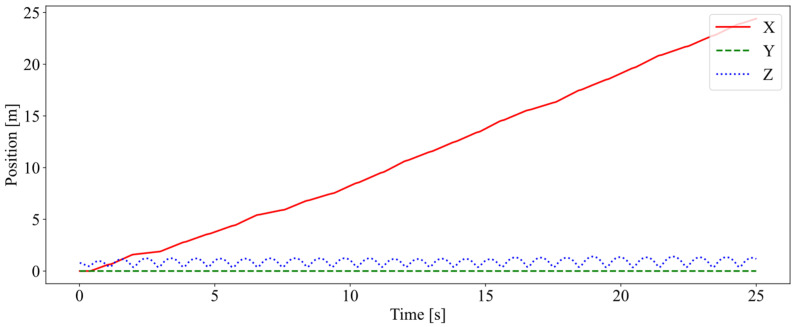
The position of the body as a function of time. X, Y, and Z denote forward, lateral, and vertical directions, respectively.

**Figure 6 biomimetics-08-00066-f006:**
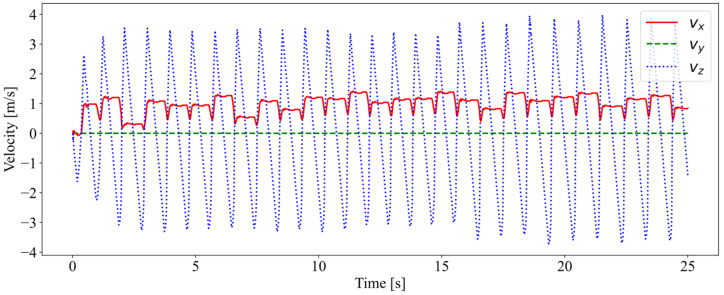
The velocity of the center of mass of the robot.

**Figure 7 biomimetics-08-00066-f007:**
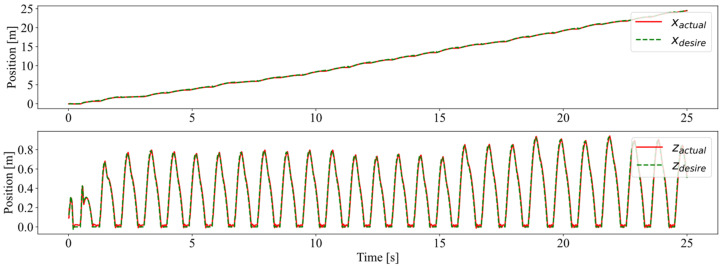
The foot position in the world coordinate system. The solid red and dashed green lines represent the actual and planned foot trajectories, respectively.

**Figure 8 biomimetics-08-00066-f008:**
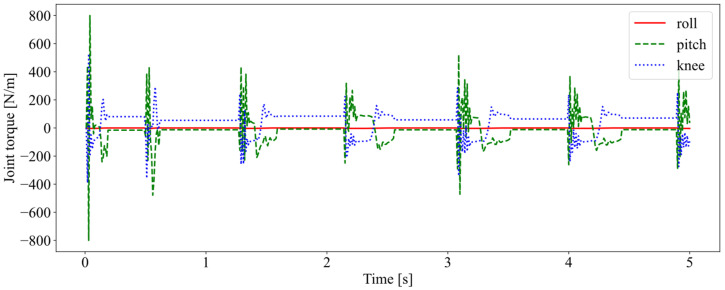
The joint torque in the first 5 s. The solid red, dashed green, and dotted blue lines represent hip roll, hip pitch, and knee pitch torques, respectively.

**Table 1 biomimetics-08-00066-t001:** Model parameters of the one-legged robot.

Parameter	Value	Definition
m_1_	6 kg	Femur link mass
m_2_	4 kg	Tibia link mass
*l* _1_	0.4 m	Femur link length
*l* _2_	0.35 m	Tibia link length
*θ* _1_	/	Hip joint roll angle
*θ* _2_	/	Hip joint pitch angle
*θ* _3_	/	Knee joint pitch angle
*l* _vir_	/	Virtual leg length
*θ* _vir_	/	Virtual leg pitch angle
*θ* _br_	/	Body roll angle
*θ* _bp_	/	Body pitch angle

**Table 2 biomimetics-08-00066-t002:** The finite-state machine for the jumping cycle.

State	Trigger Event	Action
COMPRESSION	Foot touches ground	Leg mimics spring Body attitude adjustment Gravity compensation
THRUST	Virtual leg extension	Leg mimics spring Body attitude adjustment Energy compensation Gravity compensation
SWING	Foot not touching	Servo foot trajectory Gravity compensation
LANDING	Leg stops swinging	Leg mimics spring Foot placement Gravity compensation

**Table 3 biomimetics-08-00066-t003:** The parameters set in the simulation environment.

Parameter	Symbol	Value
Virtual spring stiffness	$k_{\mathrm{ss}}$	11,000
Virtual spring damping	$k_{\mathrm{ds}}$	60
Original length of the spring	$l_{\mathrm{vir}}^{d}$	0.675
Amplitude of added force	α	2800
Speed increment coefficient	$k_{\dot{x}}$	0.01
Position gain of foot	$k_{\mathrm{pf}}$	4000
Velocity gain of foot	$k_{\mathrm{df}}$	60
Proportional gain of attitude	$k_{\mathrm{pb}}$	3000
Derivative gain of attitude	$k_{\mathrm{db}}$	80

## Data Availability

Not applicable.

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
