# Peer review of "Controlling a One-Legged Robot to Clear Obstacles by Combining the SLIP Model with Air Trajectory Planning"

_biomimetics, 2023, doi:10.3390/biomimetics8010066_

Round 1
Reviewer 1 Report
This paper proposed a 3-DoF one-legged robot, and developed a spring-loaded inverted pendulum (SLIP) model for controlling the jumping movements. Simulation analysis was carried out to evaluate this one-legged hopping robot performance in terms of jumping over obstacles by planning foot trajectory in the air based on the Bezier curve. However, the novelty and main contribution is not very clear to the reviewer, and missing of preliminary prototyping and test results also left this work an incomplete study. In addition, the paper is not well organised and presented and the English presentation improvement is required. Besides, please find my other comments below:
- In the introduction, the motivation of this paper is not clear: why this kind of leg-mechanism, what is gonna be the novelty or contribution of this work, what will be the difference of your proposed control method, etc?
- Line 56, reference is missing.
- Line 93, the specification of the robot looks unreasonable, a dimension of 0.3*0.3*0.8m but with a mass of 130 kg? Any particular reason for this specification?
- In Table 1, some parameters are not match with the symbols in Fig. 1.
- The kinematic model is not clear, where are these coordinate frame allocated (origins)? What are the directions of axes?
- In Eq. (1), what do angles of S_ij and C_ij represent?
- Line 136, why the foot trajectory is generated by a 5th order Bezier curve?
- In the simulation, it would highly recommend to list parameters set in the simulation environment, such as I_vir, Kpf, Kdf, Kp, Kd and so on
Reviewer 2 Report
The paper "Controlling a One-legged Robot to Clear Obstacles by Combin-2 ing SLIP Model with Air Trajectory Planning" by Huang and Zhang presents a jumping robot with 3-DoF and one leg.
Although the manuscript is quite interesting there are several crucial issues that need to be addressed.
Authors have to inlude information about previous one-legged robots and clarify what is new in their approach.
Some empirical result would have increased the scientific value of the work.
Authors shoudl expand the state of the art of jumping robot inspired to living organisms, and also they have to include some example from biological models.
Some relevant works that authors can include and comment in their work to improve the scientific soundness of the study
Mo, X., Ge, W., Miraglia, M., Inglese, F., Zhao, D., Stefanini, C., & Romano, D. (2020). Jumping locomotion strategies: from animals to bioinspired robots. Applied Sciences, 10(23), 8607.
Romano, D., Bloemberg, J., Tannous, M., & Stefanini, C. (2020). Impact of aging and cognitive mechanisms on high-speed motor activation patterns: evidence from an orthoptera-robot interaction. IEEE Transactions on Medical Robotics and Bionics, 2(2), 292-296.
Also, in another study, authors have modelled jumping in locusts individuals with one hindleg
Mo, X., Ge, W., Romano, D., Donati, E., Benelli, G., Dario, P., & Stefanini, C. (2019). Modelling jumping in Locusta migratoria and the influence of substrate roughness. Entomologia Generalis, 38(4), 317-332.
A deep english revision is needed.
Round 2
Reviewer 2 Report
Authors addressed almost all my comments and the manuscript is now improved.
Some minor suggestions regard the improvement of the discussion on bioispiration and more examples from the biological world shoudl be included, for instance
Mo, X., Romano, D., Miraglia, M., Ge, W., & Stefanini, C. (2020). Effect of substrates' compliance on the jumping mechanism of Locusta migratoria. Frontiers in Bioengineering and Biotechnology, 8, 661.
Gabriel, J. M. (1984). The effect of animal design on jumping performance. Journal of Zoology, 204(4), 533-539.
Also, English need a deep revision.
